# History of breastfeeding but not mode of delivery shapes the gut microbiome in childhood

Camille C. Cioffi[1], Hannah F. Tavalire[1], Jenae M. Neiderhiser[2], Brendan Bohannan[1], Leslie D. Leve[1] *

**1** University of Oregon, Eugene, OR, United States of America, **2** The Pennsylvania State University, University Park, PA, United States of America

* leve@uoregon.edu

## Abstract

### Background

The naïve neonatal gut is sensitive to early life experiences. Events during this critical developmental window may have life-long impacts on the gut microbiota. Two experiences that have been associated with variation in the gut microbiome in infancy are mode of delivery and feeding practices (eg, breastfeeding). It remains unclear whether these early experiences are responsible for microbial differences beyond toddlerhood.

### Aims

Our study examined whether mode of delivery and infant feeding practices are associated with differences in the child and adolescent microbiome.

### Design, subjects, measures

We used an adoption-sibling design to compare genetically related siblings who were reared together or apart. Gut microbiome samples were collected from 73 children ($M$ = 11 years, $SD$ = 3 years, $range$ = 3–18 years). Parents reported on child breastfeeding history, age, sex, height, and weight. Mode of delivery was collected through medical records and phone interviews.

### Results

Negative binomial mixed effects models were used to identify whether mode of delivery and feeding practices were related to differences in phylum and genus-level abundance of bacteria found in the gut of child participants. Covariates included age, sex, and body mass index. Genetic relatedness and rearing environment were accounted for as random effects. We observed a significant association between lack of breastfeeding during infancy and a greater number of the genus *Bacteroides* in stool in childhood and adolescence.

**Data Availability Statement:** All relevant data are within the manuscript and its Supporting Information files.

**Funding:** This study was supported by grants UH3 OD023389 from the Office of The Director, National Institutes of Health (PIs: Leslie Leve, Jenae Neiderhiser, Jody Ganiban), R01 DA035062 from the National Institute on Drug Abuse, the National Institute of Mental Health and OBSSR, NIH, U.S. PHS (PI: Jenae Neiderhiser), R01 HD042608 from the Eunice Kennedy Shriver National Institute of Child Health & Human Development, NIH, U.S. PHS (PI: Leslie Leve), R01 DA035062 from the National Institute on Drug Abuse, NIH, U.S. PHS (PI: Leslie Leve) and the Faculty Alumni Award from the College of Education, University of Oregon. The content is solely the responsibility of the authors and does not necessarily represent the official views of the National Institute of Child Health & Human Development or the National Institutes of Health.

**Competing interests:** The authors have declared that no competing interests exist.

## Conclusion

The absence of breastfeeding may impart lasting effects on the gut microbiome well into childhood.

## Introduction

The gut microbiome contains a vast collection of microorganisms residing in the human gastrointestinal ecosystem. The microbial composition of the human gut microbiome has been implicated as part of the etiology of both healthy and diseased states [1–3]. In the past decade, research on the interaction between the host and its microbiota has flourished. Individual variation in the human microbiome has been attributed to a variety of factors [4]. Two such factors that have been shown to be salient for predicting gut-microbiome composition during infancy are mode of delivery (MOD), specifically, whether a child was born via vaginal delivery or cesarean section [5–7], and feeding practices (FP; ie, breastfed v. formula fed) [4,8]. Although these factors have been associated with variation in the gut microbiome in infancy and toddlerhood, it is unclear whether associations persist into childhood and adolescence. Whether or not these early life experiences predict microbial composition into childhood may impact the emphasis placed on interventions for cesarean delivered (e.g., vaginal swabbing) [9] and formula fed infants (e.g., probiotic supplementation in formula) [10]. For example, interventions to supplement alterations in the microbial environment occurring from exposure to cesarean delivery and formula feeding may be less relevant to the future composition of the gut microbiome if early life experiences are unrelated to microbial composition later in childhood.

Given that the microbiome interacts with host genetics, especially in the case of gut dysbiosis [11], and is highly influenced by the host environment [12], it is important to account for differences in host genetic background and shared rearing environment. The current adoption design accounts for genetic relatedness among siblings in shared versus separate home environments using a sample of adoptees, their adoptive siblings reared together in the adoptive home, and their biological siblings reared apart from the adoptee in the biological home [13]. Using this sibling-adoption design, we examined the abundance of bacteria at the phylum and genus levels of taxonomy while controlling for known influences on the gut microbiome including body mass index, age, sex, related pairs and households [11,14] to characterize the association between early life experiences and gut microbiome composition during childhood and adolescence.

## Methods

### Participants

Participants are a subset of children from the Early Growth and Development Study, which is a prospective, longitudinal adoption study [15], and their siblings. Adopted children and their genetically related siblings reared apart or together, as well as their genetically unrelated siblings reared in the same household, were part of the subset who participated ($n$ = 73). Adopted children were placed in the home within approximately 90 days after birth. Fifty-one percent of children were female, and the average age at the time of the stool collection was 11 years old (SD = 3, range = 3–18 years). There were a total of 32 linked sibling constellations with two to six children per constellation. In terms of the rearing environment, 66% ($n$ = 48) of children

were reared in an adoptive home, and 34% (*n* = 25) were reared in the biological home. Table 1 provides information about the rearing environment for the adopted children, their genetically related siblings, unrelated siblings in the adoptive home, and other children in the birth parent home. The BMI in the sample was age corrected using the Centers for Disease Control and Prevention growth charts [16] and was, on average 20.5 (SD = 5.8). Research was approved by the University of Oregon institutional review board (protocol number: 09032013.002). Written consent was obtained for all participants. This study included children under the age of 18. Consent for child participation was obtained from the parent or guardian.

## Microbiome collection and analyses

Samples were collected from July 2016 to September 2017, in the home, using the Omnigene fecal collection kit following kit instructions (Genotek OMR-200) and returned via standard mail. Upon receipt, fecal samples were frozen at -20 degrees Celsius until they could be resuspended in a PBS buffer solution as needed and frozen at -80 degrees Celsius until DNA extraction. Metadata were collected using a survey booklet returned with the samples. The survey booklet included information about the sampling date/time, child age, sex, height, weight, and feeding practices in infancy. MoBio PowerFecal® DNA Isolation Kits were used to extract DNA from stool samples following the procedure outlined by the manufacturer. Samples and negative controls were sequenced on the Illumina HiSeq4000 sequencing platform using paired-end 150bp reads with a target sequencing depth of 50k reads per sample. Quality filtering was done in QIIME2 [17] using default settings, and the DAD2 pipeline was used to identify amplicon sequence variants (ASVs) at 100% sequence similarity from the 16S ribosomal RNA variable region V4 [18]. The sequencing depth of final, quality filtered libraries ranged from 39,523 to 84,296 reads with 143 to 469 unique ASVs identified. Alpha diversity metrics (Shannon's H, Pielou's evenness index, Faith's phylogenetic diversity index) were calculated in QIIME2. Data were rarified to 39,500 reads for subsequent analyses comparing phylum and genus level abundances [19]. We observed no effect of transportation and freezing time on variation in alpha diversity (Pearson's r = -0.04, *p* = 0.72) or sequencing depth (Pearson's *r* = -0.07, *p* = 0.56).

## Feeding practices

Parents were asked to report on whether their child was breastfed or formula fed. If parents indicated that their child was breastfed for any duration of time, they were classified as breastfed, whereas infants who were never breastfed were classified as formula fed. However, we acknowledge that infants who were not breastfed may not have consistently been formula fed. We use the term formula to include the wide variety of formula types, some of which may be created by the infant's rearing parent, rather than purchased as marketed formula.

**Table 1. Child rearing environment.**

|  | Adoptive home | Biological home | Total |
|---|---|---|---|
| Adopted child | 25 | 0 | 25 |
| Sibling genetically related to adopted child | 17 | 11 | 28 |
| Child genetically unrelated to adopted child | 6 | 14 | 20 |
| Total | 48 | 25 |  |

'Siblings genetically related to the adopted child' could include siblings with the same biological mother and father or just one biological parent in common. 'Child genetically unrelated' are children who may have also been adopted, but did not have the same biological parent as the focal adopted child from the larger study, or could be a biological child of one or both of the adoptive parents.

## Mode of delivery

MOD was collected from all 25 adoptees and from 18 biological siblings reared in the biological home from medical records. Medical records were missing for 30 children, and was thus collected by phone interview from the parent. These data collection efforts were nested within data collection efforts for the larger study.

## Covariates

Body mass index (BMI), age, and biological sex were collected in the booklet at the time of microbiome sample collection (mother report). For BMI, of the children in our study, 57% fell in the normal range (5th percentile to 85th percentile), 19% had over weight (85th to 95th percentile), 3% had underweight ($< 5$th percentile), and 21% had obesity ($> 95$th percentile). For analyses, BMI was computed using an age corrected z-score calculated based on the publicly available normalization procedures of the CDC [16]. Age was rounded to the nearest whole year, and sex was dichotomized assigning males as the reference group.

## Analyses

Microbiome count data have distinct properties such as zero-inflation and over-dispersion [20]. Thus, mixture models with a negative binomial or Poisson error distribution were considered as possible analytic approaches for examining associations between MOD and FP and microbial abundance at various levels of taxonomy. Given that the Poisson distribution assumes the mean and variance are equal, we analyzed differences between the means and variances for each taxa and consistently found the variance was at least two-fold greater than the mean for each taxa *(average variance to mean ratio* = 515.12), making the negative binomial distribution more appropriate in order to handle over-dispersion in the data and ensure proper parameter estimation [20]. Moreover, negative binomial mixture models are appropriate for microbiome data given that the microbial data in this sample are nested within the host and individuals are nested within related pairs and within households [20]. Our model included a nested random effect allowing the intercept to vary among home and family and within home [21] to account for differences in bacterial abundance due to genetic relatedness and rearing environment. Additionally, we tested whether host gut microbiome alpha diversity (mean species diversity) was associated with MOD and FP. Shannon, Pielou, and Faith are continuous indices of alpha diversity which account for relative abundance and sequencing depth in different ways [22]. We examined whether MOD and FP were associated with these three metrics using a general linear model controlling for age, sex, BMI, and genetic relatedness and rearing environment. To assess whether the total count of ASVs present in each sample was related to our variables of interest, we performed a Poisson regression. We used permutational multivariate analysis of variance (PERMANOVA) to estimate the relative contributions of MOD and FP to beta diversity (pairwise differences in microbiome diversity estimated using Bray Curtis dissimilarity). All analyses were completed in R v3.4.3. The package *glmmADMB* [23,24] was used for all mixture models. The package *vegan* was used for PERMANOVA analysis [25].

## Results

In our study's subsample of adoptees and their siblings reared in the adoptive home and siblings reared in the biological home, 69% were delivered vaginally ($n = 50$), and 21% were breastfed ($n = 15$; see Table 2). We identified 11 phyla and 96 genera within the sample of 73 participants. Relative abundance of the most common phyla and genera across individuals

**Table 2. Mode of delivery and feeding practices by rearing environment.**

|  | Adoptive home | Biological home | Total |
|---|---|---|---|
| Delivery type |  |  |  |
| Cesarean section | 19% (14) | 13% (9) | 32% (23) |
| Vaginal delivery | 46% (34) | 22% (16) | 68% (50) |
| Feeding practice |  |  |  |
| Formula fed | 55% (40) | 24% (18) | 79% (58) |
| Breastfed | 11% (8) | 10% (7) | 21% (15) |

Number of children noted in parentheses.

within each MOD and FP group is depicted in Figs 1,2, 3 and 4. Results from negative bino-mial mixture models suggest that mode of delivery was unrelated to the presence of taxa at the phylum and genus levels after accounting for false discovery rate [26]. However, FP was signif-icantly associated with abundance of the genus *Bacteroides*, as shown in Table 3 and Fig 3. Spe-cifically, when children were breastfed as infants, the expected counts of the *Bacteroides* in the child's gut microbiome were 0.46 fold those of children who were never breastfed ($p < .0001$). A box-plot with mean differences between breastfed and formula fed children on *Bacteroides* abundance is provided in Fig 5. There were no associations between any measures of alpha diversity, beta diversity, or total ASV count and MOD and FP.

## Discussion

Using negative binomial mixture models to account for over-dispersion, genetic relatedness, and the shared rearing environment, we did not find any differences in child microbiomes associated with breastfeeding and vaginal delivery at the genus and phylum levels, except for the genus *Bacteroides*. Specifically, we identified a greater abundance of *Bacteroides* in the gut microbiomes of children who were not breastfed as infants compared to infants who were breastfed. This finding suggests that the direct effects of FP and MOD on the gut microbiome

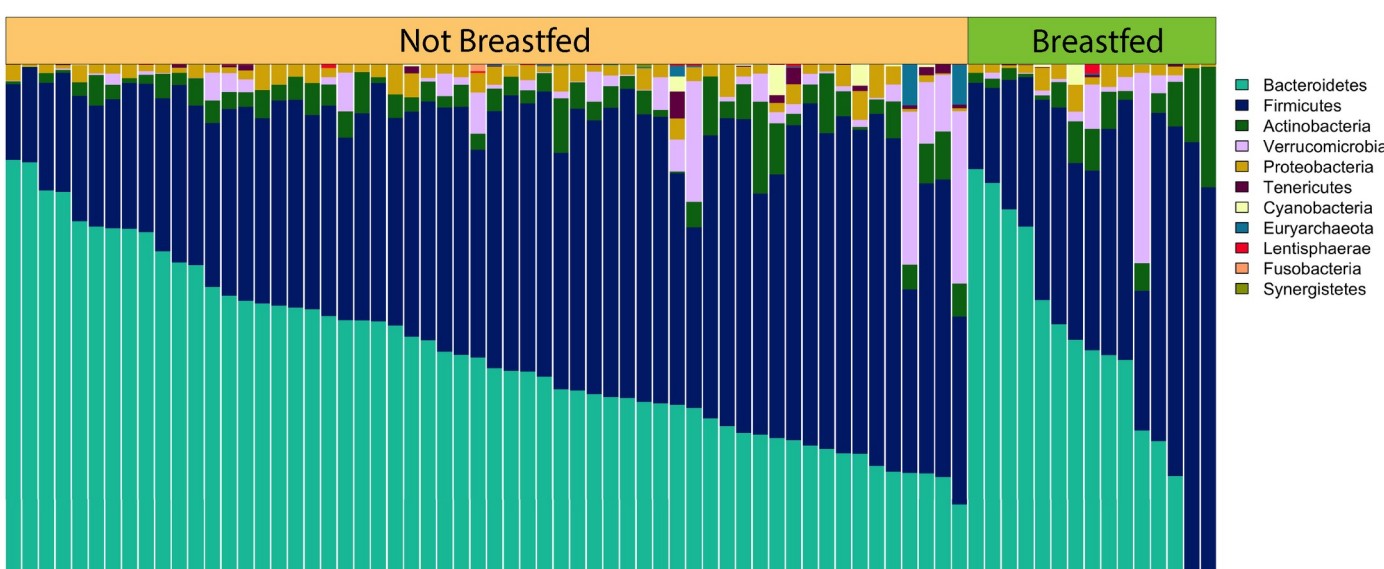

**Fig 1. Relative abundance of the most common phyla by feeding practice (FP).** Each vertical bar represents an individual. FP group is designated across the top of the figure. Samples are ordered within each group by descending Bacteroidetes abundance.

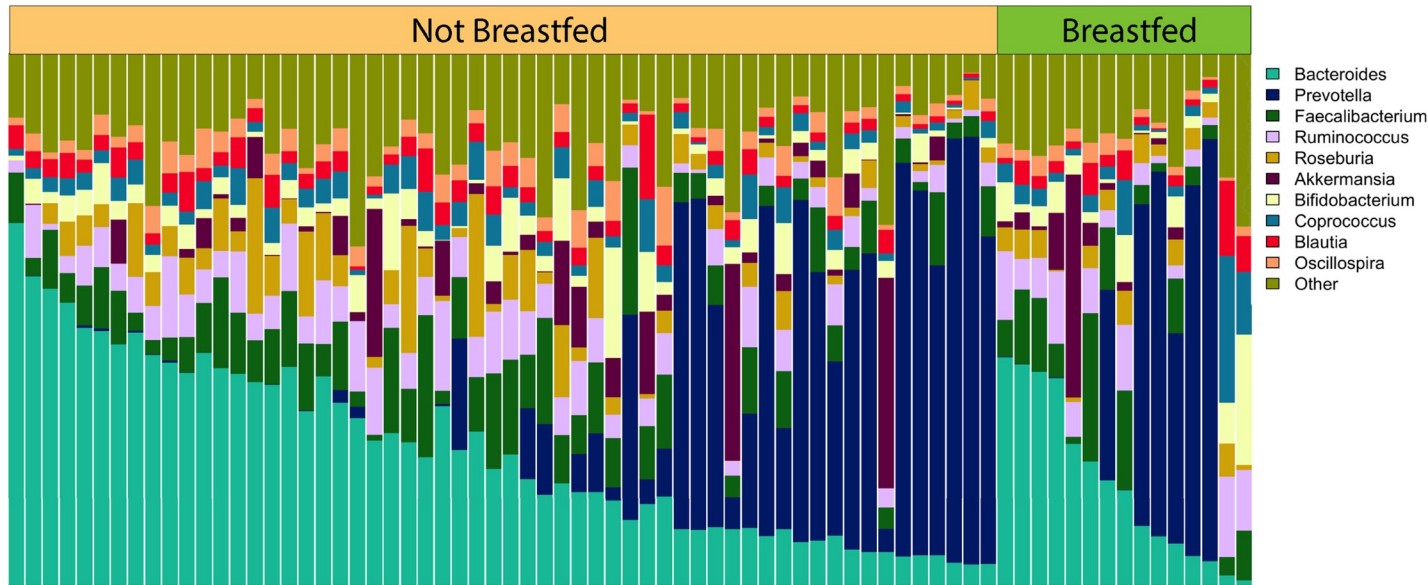

**Fig 2. Relative abundance of the most common genera by feeding practice (FP).** Each vertical bar represents an individual. FP group is designated across the top of the figure. Samples are ordered within each group by descending *Bacteroides* abundance.

may become obsolete in childhood except for the influence of FP on the relative abundance of *Bacteroides*. This finding is meaningful given that *Bacteroides* is a predominate genus in the human gut microbiome (in fact, *Bacteroides* live and grow exclusively in the mammalian digestive tract) and are a known driver of gut maturation and diversity [27,28]. Moreover, *Bacteroides* have been shown to improve their host's ability to fight infections by enteric pathogens and more generally improve immune tolerance [29,30]. However, *Bacteroides* have also been associated with problematic outcomes in the host.

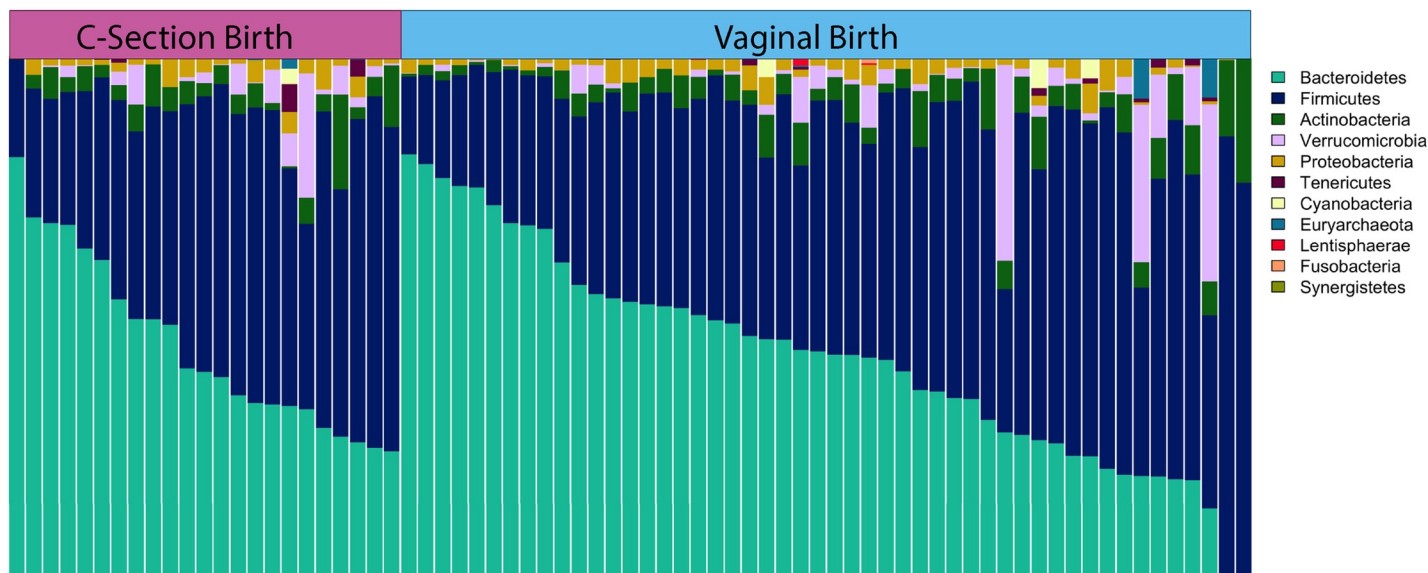

**Fig 3. Relative abundance of the most common phyla by mode of delivery (MOD).** Each vertical bar represents an individual. MOD group is designated across the top of the figure. Samples are ordered within each group by descending Bacteroidetes.

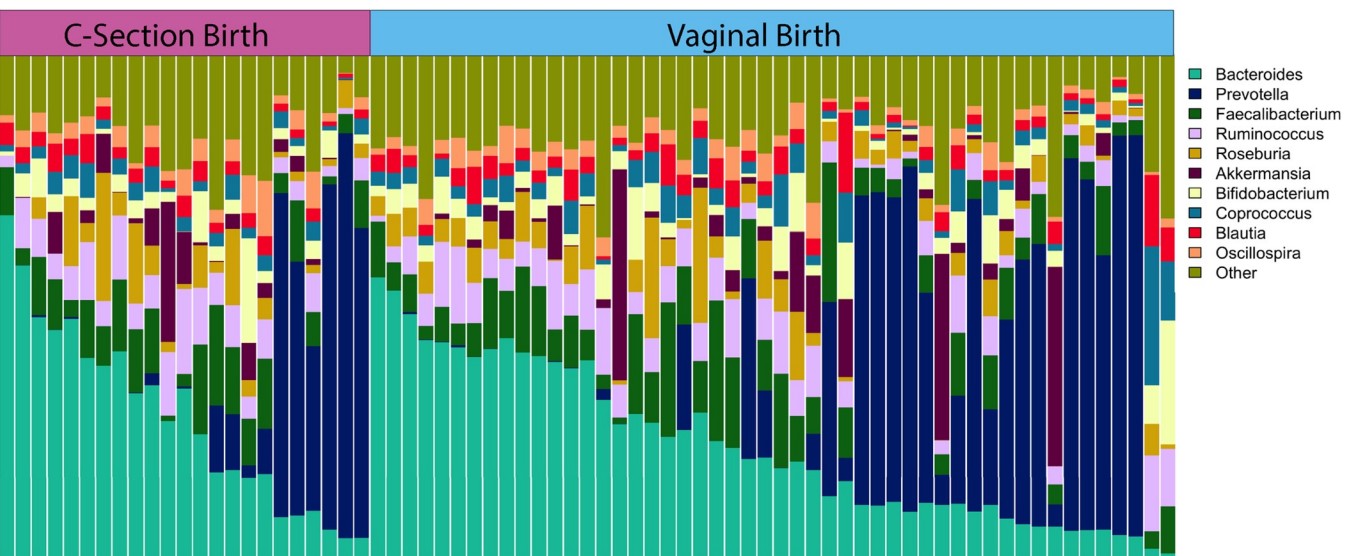

**Fig 4. Relative abundance of the most common genera by mode of delivery (MOD).** Each vertical bar represents an individual. MOD group is designated across the top of the figure. Samples are ordered within each group by descending *Bacteroides* abundance.

## Implications

Most research on the association between MOD and microbiome composition has been completed within the first three years of life, likely because it has been proposed that the gut microbiome converges to an adult-like state between the ages of 3 and 5 and remains stable in later life [31]. Previous research found that the microbiome maintained individual uniqueness but converged towards a relatively stable, adult-like trajectory after the age of 3 [28,32]. However, recent studies in older children suggest that the microbiome changes throughout childhood to support shifting developmental needs [28]. Our study suggests that changes in the microbiome into adolescence, may erase many of the effects of early life experiences on microbiome composition.

**Feeding practices.** Prior studies of infants and toddlers have found that breastfeeding is associated with greater abundances of the genera *Bifidobacteria*, *Streptococcus*, *Bacteroides*, *Firmicutes*, *Lactobacilli-EnterocoiI* [10,33–35] and the phyla *Firmicutes*, *Bacteroidetes*, and *Actinobaceria* [36]. Formula feeding has been associated with a greater abundances of *Clostridium*, *Streptococcus*, *Enterococcus*, and *Veillonella* [10,33,34] and the phylum *Proteobaceria* [36]. Thus, it was surprising that FP were related only to the abundance of *Bacteroides* in our sample

**Table 3. Negative binomial mixture model of mode of delivery and breastfeeding for *Bacteroides*.**

| Fixed effects | Estimate | SE | z | p |
|---|---|---|---|---|
| Age | 1.01 | .02 | .83 | .41 |
| Sex | 0.87 | .09 | -1.53 | .13 |
| BMI | 0.98 | .05 | -0.36 | .72 |
| Vaginal birth | 0.87 | .11 | -1.19 | .24 |
| Breastfed | 0.46 | .20 | -3.91 | < .0001 |
| Random effects | Variance | SD | | |
| Environment | .22 | .47 | | |
| Genetic relatedness | .22 | .47 | | |

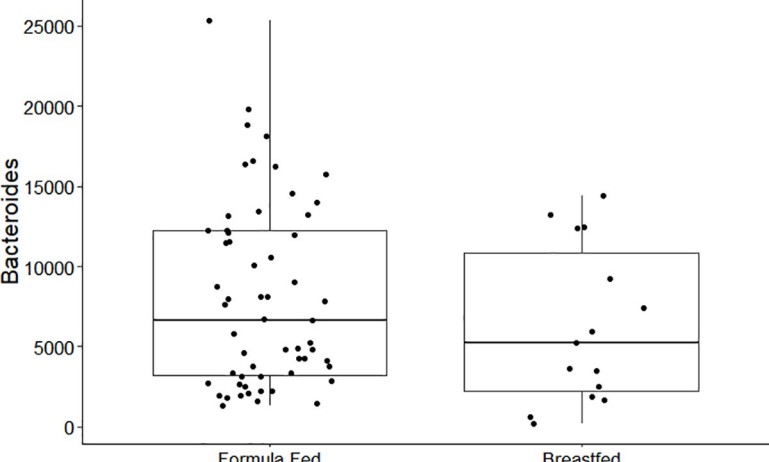

**Fig 5. Boxplot of mean differences between formula feeding and breastfeeding for the genus *Bacteroides*.**

(i.e., children who were formula fed harbored a greater abundance of *Bacteroides* in childhood compared to their breastfed counterparts). *Bacteroides* a predominate genus in the human microbiome, is a known driver of gut maturation and diversity [27,28], and has been shown to improve host resistance to pathogen colonization and improves human immune tolerance [29,30]. It should be noted that *Bacteroides* may, in some cases, cause harm to the host. For example, in the presence of inflammation, an abundance of specific *Bacteroides* species can enhance the pathogenicity of enterohemorrahagic *E. Coli* during inflammation [37,38] greater abundance of *Bacteroides* has also been linked to the prevalence of type 1 diabetes [39], and *Bacteroides* may cause infection if they escapes from the gut, potentially leading to septicemia [27].

Since some research suggests that *Bacteroides* are more abundant in breastfed infants [34], we might have expected that children with a history of breastfeeding rather than formula feeding would have harbored a greater abundance of *Bacteroides*. Instead we found that *Bacteroides* were more abundant for children with a history of formula feeding. This may be a result of early exposure to foods other than breast milk for infants who were formula fed. For example, *Bacteroides* are found in higher abundance in the gut microbiome of as children begin consuming solid foods [40], and higher abundance of *Bacteroides* is associated with a more mature (i.e., more adult-like) gut microbiome [28]. More research is needed to understand the associations between a higher abundance of *Bacteroides* and child outcomes, especially as is related to diabetes [39].

**Mode of delivery.**   It is surprising that there was no evidence of differences in diversity or relative abundance of taxa in the gut microbiome associated with MOD in the current study, especially in light of differences observed in prior studies in the gut microbiota of infants and young children who were delivered by cesarean section and infants birthed vaginally [28]. Studies comparing the composition of the microbiome in infancy relative to MOD have revealed higher abundance of several genera in infants born vaginally compared to infants born by cesarean section [4]. For example, the microbes belonging to the *Bacteroides*, *Bifidobacterium*, *Lactobacillus*, *Prevotella*, *and Snethia* genera have all been found to be more abundant in infants delivered vaginally [5–7,41–44], whereas microbes belonging to the *Blautia*, *Prevotella*, *Staphylococcus*, *Corynebacterium*, *Propionibacterium*, *and Clostridium* genera are more abundant in the gut of infants delivered by cesarean section [7,41,42,44,45]. Of particular importance is the microbial presence and abundance of *Clostridium difficile* in cesarean

 

section born infants, which is associated with health challenges including diarrhea and food poisoning [46]. Higher abundance of *Clostridium* has also been found in 7-year old children with a history of cesarean birth [47]. Interestingly, this same study did not find differences in microbiome diversity, the presence of *Bacteroides*, *Bifidobacterium*, or *Lactobacillus*. The absence of an association between MOD and the child gut microbiome in the current sample highlights the importance of observing the microbiome into later childhood to identify the persistence of microbial alterations as a result of differences in MOD. Our findings corroborate recent evidence suggesting that altering the microbiome of cesarean delivered infants to resemble vaginally delivered infants may not be a useful mechanism for improving individual host fitness [48]. For instance, research finding that cesarean section is associated with a higher incidence of problematic outcomes, such autoimmune diseases [6], has given rise to the practice of vaginal seeding for infants born by cesarean section [9]. Recent opinion has challenged this practice based on the dearth of well-designed studies on the association between cesarean section, microbiome composition and disease outcomes [48].

**Additional considerations.** Previous research reported effects of MOD and FP on the development of the infant microbiome; our research suggests that most of these effects may not be associated with the gut microbiome in childhood and adolescence. Additionally, potential environmental confounds may exist which are more salient predictors of microbial composition than early life experiences, such as specific aspects of the rearing environment which we have not accounted for in our study. It may also be that long term impacts of FP and MOD on gut microbial composition vary by geographical location, and samples in the current study were collected across a broad geographic range across the United States, such that comparisons of specific geographical regions were not feasible. Microbiota vary across geographic locations as a function of diet, cultural practices, and living situations [14,49]. Longitudinal studies of microbial composition in response to FP and MOD are needed in order to assess both the short- and long-term effects of early life experiences on the child and adolescent gut microbiome.

## Strengths and limitations

Our study aimed to address the dearth of information on the effects of FP and MOD on gut microbiome composition in later childhood. There are several strengths to this approach, including the use of negative binomial mixture models to account for over-dispersion and genetic relatedness among siblings and home rearing environment. This is the first study, to our knowledge, to apply these techniques using a sibling-adoption design to account for rearing environment and genetic relatedness. Our study is also one of few that looks beyond the first four years of life to assess associations between FP and MOD and gut microbial composition [47,50,51] Some limitations to consider are that the current study had diminishing power to detect statistically significant associations in a sample size of $n = 73$. Larger studies must be completed in order to confirm our results, although our sample size was sufficient to detect an effect size of 0.4 or larger. Second, because of the use of an adoption sample, a lower proportion of children were ever breastfed compared to the general population of the United States (79%) [52], which increased variability in FP but limited our ability to explore differences in duration and exclusivity of breastfeeding and may limit generalizability and limited our ability to examine duration of breastfeeding and the use of breastmilk and formula simultaneously. Moreover, our data did not capture whether breastmilk came from other sources, such as friends, family, or community support breastmilk networks. Retrospective reports of feeding practices may also be inaccurate. Additionally, this study was unable to control for known influences of the gut microbiome, such as diet and antibiotic use [53,54]. Thus, the exclusion of these variables from our analytic models could have affected the results.

## Conclusion

This work highlights that the effects of two early life experiences (MOD and FP), while important, do not necessarily impact the long-term development of the child gut microbiome. However, early feeding was related to the abundance of the genus *Bacteroides* in later childhood and adolescence, a known marker of gut maturity and diversity that provides benefits to the human immune system [27] but may also cause problems in the host [37,38,39]. This finding implies that early feeding may impart lasting effects on the gut microbiome well into childhood.

## Supporting information

**S1 Dataset. Child characteristics and microbiome data.**
(CSV)

## Acknowledgments

We would like to thank Diana Christie for sequencing our Microbiome data, the Early Growth and Development Study team who collected data, and all of the children and parents who participated in our study.

## Author Contributions

**Conceptualization:** Brendan Bohannan, Leslie D. Leve.

**Data curation:** Hannah F. Tavalire, Brendan Bohannan, Leslie D. Leve.

**Formal analysis:** Camille C. Cioffi, Hannah F. Tavalire.

**Funding acquisition:** Jenae M. Neiderhiser, Brendan Bohannan, Leslie D. Leve.

**Investigation:** Jenae M. Neiderhiser, Brendan Bohannan, Leslie D. Leve.

**Methodology:** Camille C. Cioffi, Hannah F. Tavalire.

**Project administration:** Jenae M. Neiderhiser, Brendan Bohannan, Leslie D. Leve.

**Resources:** Leslie D. Leve.

**Supervision:** Hannah F. Tavalire, Jenae M. Neiderhiser, Brendan Bohannan, Leslie D. Leve.

**Validation:** Hannah F. Tavalire.

**Visualization:** Camille C. Cioffi, Hannah F. Tavalire.

**Writing – original draft:** Camille C. Cioffi, Hannah F. Tavalire.

**Writing – review & editing:** Camille C. Cioffi, Hannah F. Tavalire, Jenae M. Neiderhiser, Brendan Bohannan, Leslie D. Leve.

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
