## [Decision Letter · Decision Letter 0]

6 Mar 2020

PONE-D-20-02497

History of breastfeeding but not mode of delivery shapes the gut microbiome in childhood

PLOS ONE

Dear Dr Leve,

Thank you for submitting your manuscript to PLOS ONE. After careful consideration, we feel that it has merit but does not fully meet PLOS ONE’s publication criteria as it currently stands. Therefore, we invite you to submit a revised version of the manuscript that addresses the points raised during the review process.

We would appreciate receiving your revised manuscript by Apr 20 2020 11:59PM. To enhance the reproducibility of your results, we recommend that if applicable you deposit your laboratory protocols in protocols.io, where a protocol can be assigned its own identifier (DOI) such that it can be cited independently in the future. For instructions see: http://journals.plos.org/plosone/s/submission-guidelines#loc-laboratory-protocols

We look forward to receiving your revised manuscript.

Kind regards,

Corrie Whisner

Academic Editor

PLOS ONE

Journal Requirements:

2. Please provide additional details regarding participant consent. In the ethics statement in the Methods and online submission information, please ensure that you have specified, if your study included minors under age 18, whether you obtained consent from parents or guardians.

3. To comply with PLOS ONE submission guidelines please deposit your sequencing data in a publicly available repository (you can find a list of repositories in the link here : https://journals.plos.org/plosone/s/data-availability#loc-recommended-repositories).

4. Please include in your Methods section the date ranges over which you recruited participants to this study.

5. Please ensure that you refer to Figure 4 and 5 in your text as, if accepted, production will need this reference to link the reader to the figure.

Reviewers' comments:

Reviewer's Responses to Questions

**Comments to the Author**

1. Is the manuscript technically sound, and do the data support the conclusions?

Reviewer #1: Partly

Reviewer #2: Yes

2. Has the statistical analysis been performed appropriately and rigorously? 

Reviewer #1: I Don't Know

Reviewer #2: Yes

3. Have the authors made all data underlying the findings in their manuscript fully available?

Reviewer #1: Yes

Reviewer #2: Yes

4. Is the manuscript presented in an intelligible fashion and written in standard English?

Reviewer #1: Yes

Reviewer #2: Yes

5. Review Comments to the Author

Reviewer #1: This paper adds to the literature on the effects of delivery mode and infant feeding on the gut microbiota by examining whether differences in diversity and bacterial composition can be seen in children aged 3-18 years by delivery and breast v formula feeding. While this is an important question, the relatively small sample size, with a wide range of child ages 3-18, and the lack of other control variables limit the generalizability of the study. Further, the authors make several strong claims that do not appear to be supported by the data and/or results.

First, the authors claim that this is the first paper to explore whether the effects of these early childhood exposures persist past age 4 (for example, pg 3 lines 44-45, pg 13, line 271). At least three other papers have examined these associations after age 4: Salminen S, Gibson GR, McCartney AL, et a influence of mode of delivery on gut microbiota composition in seven year old children. Gut 2004;53:1388-1389; Thompson, A. L., Houck, K. M., & Jahnke, J. R. (2019). Pathways linking caesarean delivery to early health in a dual burden context: Immune development and the gut microbiome in infants and children from Galápagos, Ecuador. American Journal of Human Biology, 31(2), e23219; and Nagpal R, Yamashiro Y: Gut Microbiota Composition in Healthy Japanese Infants and Young Adults Born by C-Section. Ann Nutr Metab 2018;73(suppl 3):4-11. doi: 10.1159/000490841

Second, the age range of the participants is quite wide and, while the authors control for age in their multilevel models of the factors associated with Bacteroides abundance it would be helpful to see whether diversity or any other measures differ by age group. Such an analysis would provide more support for their conclusion on page 10 that: “our study suggests that changes in the microbiome occur well into adolescence…” As currently described, this cross-sectional analysis doesn’t provide evidence that the microbiome changes in childhood and adolescence, just that there are differences in the colonization that may be associated with infant feeding practices. A similar wording issue is on pg 12, line 251 where the authors describe the effects as “diminishing over time.”

Similarly, the authors should be more explicit about the limitations/potential bias in their sample data. How likely are maternal reports of child size, breastfeeding and birth type to be accurate given the mixed history of the sample (some biological children, some adoptive children with likely varying ages at adoption)? Further, how likely are other variables to have influenced their findings (i.e. timing of complementary feeding, antibiotic use, etc) given the lack of other infant or childhood exposures?

The authors don’t appear to make use of the adoption-sibling design of the study other than controlling for clustering by household and differing genetic relatedness. This seems like a lost opportunity to make this study unique in the literature by providing an examination of how household environment may interact with early birth/feeding exposures to shape child microbiota.

A few more minor concerns:

Pg 9, line 185: “latent” seems to be the wrong word

Pg 13, lines 276: It is not clear that the authors are referring to breastfeeding initiation in the US (79%) compared to other measures (i.e. duration at 3 months, ebf at 6 months). The fact that the authors only collected “ever breastfed” appears to be the more limiting factor for looking at ebf or duration rather than the lower breastfeeding initiation prevalence than the US population.

Reviewer #2: There are several areas that can be improved with additional clarification.

1. Do not assume that the children that were not breastfed were formula fed, as that gives the impression of commercial formula use. Parents may have used other types of liquids to feed their children, or to have created their own types of 'baby milk'. I recommend terms such as breastfed and non-breastfed.

2. The discussion of the participants is confusing. A table displaying the children who were adopted or biological and reared with birth parents or adoptive parents would be easier to understand than the text. The term 'relatedness' is also not clarified as it could mean 'related through shared genetics' or 'related through being members of the same family'.

3. BMI needs to be stated as BMI percentile. A BMI of 20 does not mean anything to the reader since the children ranged in age from 3 to 18. It would be more helpful to know if the BMI% is at 50th% or 90th%.

4. Knowing the child's typical diet at the age when the fecal sample was collected would be useful. A food frequency questionnaire would have been a beneficial instrument for knowing if the child had a predominantly meat heavy or vegetable heavy diet, which can affect the gut microbiome.

5. It would be helpful to know the age of adoption. Some adoptees may have been initially breastfed. Cannot assume that if adopted, must have not been breastfed.

6. Acknowledge in additional considerations that knowing the current diet (important co-variate) could also contribute to analysis of these results.

6. PLOS authors have the option to publish the peer review history of their article (what does this mean?). If published, this will include your full peer review and any attached files.

Reviewer #1: No

Reviewer #2: No

---

## [Author Response · Author response to Decision Letter 0]

9 Apr 2020

Dear Dr. Whisner,

Thank you for the opportunity to revise our manuscript entitled, “History of breastfeeding but not mode of delivery shapes the gut microbiome in childhood”. We have reviewed the comments raised by you and reviewers and have provided our responses along with a tracked-changes version of the manuscript. Page numbers correspond with the tracked changes version of the manuscript.

Academic Editor

We have reviewed our manuscript and file names to ensure consistency with PLOS ONE’s style requirements.

2. Please provide additional details regarding participant consent. In the ethics statement in the Methods and online submission information, please ensure that you have specified, if your study included minors under age 18, whether you obtained consent from parents or guardians.

We have provided additional details about participant consent on page 5.

3. To comply with PLOS ONE submission guidelines please deposit your sequencing data in a publicly available repository (you can find a list of repositories in the link here : https://journals.plos.org/plosone/s/data-availability#loc-recommended-repositories).

In line with PLOS ONE submission guidelines, data are provided in the supporting information files.

4. Please include in your Methods section the date ranges over which you recruited participants to this study.

We have included the date ranges for data collection on page 5 (from July 2016 to September 2017).

5. Please ensure that you refer to Figure 4 and 5 in your text as, if accepted, production will need this reference to link the reader to the figure.

Figures 4 and 5 are now referenced in text on pages 7 and 8. 

We have provided a caption for our supporting information after the acknowledgements on page 14. There is one supporting information file which is the dataset for this study.

Reviewer #1

1. First, the authors claim that this is the first paper to explore whether the effects of these early childhood exposures persist past age 4 (for example, pg 3 lines 44-45, pg 13, line 271). At least three other papers have examined these associations after age 4: Salminen S, Gibson GR, McCartney AL, et a influence of mode of delivery on gut microbiota composition in seven year old children. Gut 2004;53:1388-1389; Thompson, A. L., Houck, K. M., & Jahnke, J. R. (2019). Pathways linking caesarean delivery to early health in a dual burden context: Immune development and the gut microbiome in infants and children from Galápagos, Ecuador. American Journal of Human Biology, 31(2), e23219; and Nagpal R, Yamashiro Y: Gut Microbiota Composition in Healthy Japanese Infants and Young Adults Born by C-Section. Ann Nutr Metab 2018;73(suppl 3):4-11. doi: 10.1159/000490841

We have clarified that we do not believe we are the first to examine associations between early life experiences and child and adolescent microbiome composition and have included the appropriate citations on pages 3 and 13. We did not change text on page 13 because we believe this is the first published paper to utilize negative binomial mixture models and an adoption design to examine associations between early life experiences and the child/adolescent gut microbiome. We have added the citations the reviewer provided on page 13.

2. The age range of the participants is quite wide and, while the authors control for age in their multilevel models of the factors associated with Bacteroides abundance it would be helpful to see whether diversity or any other measures differ by age group. Such an analysis would provide more support for their conclusion on page 10 that: "our study suggests that changes in the microbiome occur well into adolescence..." As currently described, this cross-sectional analysis doesn't provide evidence that the microbiome changes in childhood and adolescence, just that there are differences in the colonization that may be associated with infant feeding practices. A similar wording issue is on pg 12, line 251 where the authors describe the effects as "diminishing over time."

We agree with the reviewer’s comment that our study could not address changes over time. Since associating age with the gut microbiome composition was not the focus of this study, we have removed language on pages 10 and 12 that infer change over time. 

3. Similarly, the authors should be more explicit about the limitations/potential bias in their sample data. How likely are maternal reports of child size, breastfeeding and birth type to be accurate given the mixed history of the sample (some biological children, some adoptive children with likely varying ages at adoption)? Further, how likely are other variables to have influenced their findings (i.e. timing of complementary feeding, antibiotic use, etc) given the lack of other infant or childhood exposures?

We have provided additional information about the current study limitations on page 14. Text appears as follows: 

“Retrospective reports of feeding practices may also be inaccurate. Additionally, this study was unable to control for known influences of the gut microbiome, such as diet and antibiotic use (53,54). Thus, the exclusion of these variables from our analytic models could have affected the results.”

4. The authors don't appear to make use of the adoption-sibling design of the study other than controlling for clustering by household and differing genetic relatedness. This seems like a lost opportunity to make this study unique in the literature by providing an examination of how household environment may interact with early birth/feeding exposures to shape child microbiota.

Since very few, if any, adoptees were breastfed, we do not have the equal representation needed in a crossed factorial design and thus are unable to run statistical tests for moderation by household environment. However, we would like to point the reviewer to Tavalire et al. (citation 13) which is currently under review at Science Advances and utilizes the adoption design to directly measure the effects of genetics versus the environment on microbiome composition.

A few more minor concerns:

Pg 9, line 185: "latent" seems to be the wrong word

Pg 13, lines 276: It is not clear that the authors are referring to breastfeeding initiation in the US (79%) compared to other measures (i.e. duration at 3 months, ebf at 6 months). The fact that the authors only collected "ever breastfed" appears to be the more limiting factor for looking at ebf or duration rather than the lower breastfeeding initiation prevalence than the US population.

We have addressed the minor concern highlighted by reviewer 1 on page 9. As noted on page 6, parents indicated the duration of time their child was breastfed but the responses were collapsed due to the low breastfeeding prevalence in our sample. This is also a limitation of our study that has been addressed on page 14 with the following text:

“Second, because of the use of an adoption sample, a lower proportion of children were breastfed compared to the general population of the United States (79%) (52), which increased variability in FP but limited our ability to explore differences in duration and exclusivity of breastfeeding and may limit generalizability and limited our ability to examine duration of breastfeeding and the use of breastmilk and formula simultaneously.”

Reviewer #2

1. Do not assume that the children that were not breastfed were formula fed, as that gives the impression of commercial formula use. Parents may have used other types of liquids to feed their children, or to have created their own types of 'baby milk'. I recommend terms such as breastfed and non-breastfed.

We acknowledge that infants who were not breastfed may not have been formula fed on page 5. We retained the term formula fed for consistency throughout the manuscript because using non-breastfed made some text more difficult to interpret. However, we clarify the definition of formula to include formula that may have been created or formulated by the infant’s rearing parent, rather than purchased as a marketed formula, to address the reviewer’s concerns. We used the following text:

“Parents were asked to report on whether their child was breastfed or formula fed. If parents indicated that their child was breastfed for any duration of time, they were classified as breastfed, whereas infants who were never breastfed were classified as formula fed. However, we acknowledge that infants who were not breastfed may not have consistently been formula fed. We use the term formula to include the wide variety of formula types, some of which may be created by the infant’s rearing parent, rather than purchased as marketed formula.”

2. The discussion of the participants is confusing. A table displaying the children who were adopted or biological and reared with birth parents or adoptive parents would be easier to understand than the text. The term 'relatedness' is also not clarified as it could mean 'related through shared genetics' or 'related through being members of the same family'.

We have provided a table on page 5 to clarify each group of children. We have clarified the term “relatedness” to say, “genetic relatedness”, throughout the manuscript.

3. BMI needs to be stated as BMI percentile. A BMI of 20 does not mean anything to the reader since the children ranged in age from 3 to 18. It would be more helpful to know if the BMI% is at 50th% or 90th%.

We have clarified that BMI was corrected for age using the CDC growth charts on page 4. The use of the CDC age-corrected growth charts makes the BMI directly comparable between ages.

4. Knowing the child's typical diet at the age when the fecal sample was collected would be useful. A food frequency questionnaire would have been a beneficial instrument for knowing if the child had a predominantly meat heavy or vegetable heavy diet, which can affect the gut microbiome.

We agree that child’s diet is an important predictor of microbiome composition and have noted this as a limitation of our study on page 14.

5. It would be helpful to know the age of adoption. Some adoptees may have been initially breastfed. Cannot assume that if adopted, must have not been breastfed.

Adoptions occurred, on average 3 days after birth. We have clarified this information on page 4. We agree that we cannot assume that children who were adopted were not breastfed and have clarified this on page 13. 

6. Acknowledge in additional considerations that knowing the current diet (important co-variate) could also contribute to analysis of these results.

We have acknowledged this limitation on page 14.

Thank you again for the opportunity to revise our manuscript. 

Sincerely,

Camille C. Cioffi

---

## [Decision Letter · Decision Letter 1]

27 May 2020

PONE-D-20-02497R1

History of breastfeeding but not mode of delivery shapes the gut microbiome in childhood

PLOS ONE

Dear Dr. Leve,

Thank you for submitting your manuscript to PLOS ONE. After careful consideration, we feel that it has merit but does not fully meet PLOS ONE’s publication criteria as it currently stands. Therefore, we invite you to submit a revised version of the manuscript that addresses the points raised during the review process.

In revising your manuscript, please pay special attention to the comments provided by Reviewer 2, especially the need to present body mass index as a percentile instead of in kg/m^2. Please see the bulleted list of important changes I noted when reviewing the paper and the comments from reviewers:

For children, the World Health Organization suggests that this approach to reporting child size and body weight status are more accurate given that they are in a period of more intense growth. This is a best practice in both clinical and research settings, and should therefore be carried out in this paper before it will be deemed acceptable for publication.The paper suggests that formula feeding is fine and may be beneficial given the promotion of gut maturation ia Bacteroides. This message needs to be softened throughout the manuscript as many health agencies support breastfeeding as the gold standard for gut and systemic child health. Please revise the manuscript to ensure that formula feeding is not viewed as equal or superior to breastfeeding.

We look forward to receiving your revised manuscript.

Kind regards,

Corrie Whisner

Academic Editor

PLOS ONE

Reviewers' comments:

Reviewer's Responses to Questions

**Comments to the Author**

1. If the authors have adequately addressed your comments raised in a previous round of review and you feel that this manuscript is now acceptable for publication, you may indicate that here to bypass the “Comments to the Author” section, enter your conflict of interest statement in the “Confidential to Editor” section, and submit your "Accept" recommendation.

Reviewer #1: All comments have been addressed

Reviewer #2: (No Response)

2. Is the manuscript technically sound, and do the data support the conclusions?

Reviewer #1: (No Response)

Reviewer #2: Partly

3. Has the statistical analysis been performed appropriately and rigorously? 

Reviewer #1: (No Response)

Reviewer #2: Yes

4. Have the authors made all data underlying the findings in their manuscript fully available?

Reviewer #1: (No Response)

Reviewer #2: Yes

5. Is the manuscript presented in an intelligible fashion and written in standard English?

Reviewer #1: (No Response)

Reviewer #2: Yes

6. Review Comments to the Author

Reviewer #1: (No Response)

Reviewer #2: I read your rationale why you used BMI only. But it lacks clarity on the predominant body size of the children. Please use BMI% for the BMI of 20 to make sense to readers. I know it's age corrected but it still doesn't tell me of the child's weight status; normal (5% to 85%) over/under weight, (85%-<5th%) obese >95%.

The child's rearing environment is confusing, especially Table 1. Does the box 'sibling genetically related and adoptive home mean that of the 25 adopted children, there were 17 genetically sibs who were reared together in an adoptive home? The the box 'child genetically unrelated to adopted child and biological home mean that in the biological home, 14 of children were there who were not related to the adoptees? For example, step-siblings and not half-siblings?

For feeding practices, did you also measure if the adoptive parents used breast milk from a milk bank or from friends/family/community support breast milk networks? That practice is becoming more common and could have occurred in the younger children in your sample.

Was the delivery mode measured by questions from adoptive or birth mother when records were not available?

In the results section, was the 21% breastfed combined from both genetically related and non-genetically related sibs?

On line 195, add 'gut' between human and microbiome.

Line 222 Mention also that bacteriodes can also cause serious infections if escape outside the gut, potentially leading to septicemia too.

Line 282 The percentage of breastfed varies by age of infant. Do you mean ever breastfed, exclusively breastfed, or breastfed to 6 months? more specificity needed.

Line 292 remove formula; it may be interpreted as 'better than breastfeeding' and that's not the approach supported by WHO or multiple health agencies.

7. PLOS authors have the option to publish the peer review history of their article (what does this mean?). If published, this will include your full peer review and any attached files.

Reviewer #1: No

Reviewer #2: No

---

## [Author Response · Author response to Decision Letter 1]

2 Jun 2020

Dear Dr. Whisner,

Thank you for the opportunity to revise our manuscript entitled, “History of breastfeeding but not mode of delivery shapes the gut microbiome in childhood”. We have reviewed the comments raised by you and reviewers and have provided our responses along with a tracked-changes version of the manuscript. 

Editor

1. For children, the World Health Organization suggests that this approach to reporting child size and body weight status are more accurate given that they are in a period of more intense growth. This is a best practice in both clinical and research settings, and should therefore be carried out in this paper before it will be deemed acceptable for publication.

We now report body size as body mass index as a percentile on line 122.

2. The paper suggests that formula feeding is fine and may be beneficial given the promotion of gut maturation in Bacteroides. This message needs to be softened throughout the manuscript as many health agencies support breastfeeding as the gold standard for gut and systemic child health. Please revise the manuscript to ensure that formula feeding is not viewed as equal or superior to breastfeeding.

We have softened the language throughout the manuscript to avoid suggesting that formula feeding may be equal or superior to breastfeeding, see lines 203, 240, and 305 in particular.

Reviewer #1:

No Comments 

Reviewer #2: 

1. I read your rationale why you used BMI only. But it lacks clarity on the predominant body size of the children. Please use BMI% for the BMI of 20 to make sense to readers. I know it's age corrected but it still doesn't tell me of the child's weight status; normal (5% to 85%) over/under weight, (85%-<5th%) obese >95%.

As noted above in the response to the Editor, we now report body size as body mass index as a percentile on line 122.

2. The child's rearing environment is confusing, especially Table 1. Does the box 'sibling genetically related and adoptive home mean that of the 25 adopted children, there were 17 genetically sibs who were reared together in an adoptive home? The box 'child genetically unrelated to adopted child and biological home mean that in the biological home, 14 of children were there who were not related to the adoptees? For example, step-siblings and not half-siblings?

The reviewer’s interpretation of Table 1 is accurate. We have added a note in the Table to clarify this for readers.

3. For feeding practices, did you also measure if the adoptive parents used breast milk from a milk bank or from friends/family/community support breast milk networks? That practice is becoming more common and could have occurred in the younger children in your sample.

Unfortunately, we did not measure if the adoptive parents used breast milk or from friends/family/community support breast milk networks. We have noted this limitation on line 286.

4. Was the delivery mode measured by questions from adoptive or birth mother when records were not available?

Delivery mode was measured by questions from adoptive or birth mother when records were not available. This information is provided on line 112. 

5. In the results section, was the 21% breastfed combined from both genetically related and non-genetically related sibs?

The 21% was across all children in our study. We now provide clarification on line 147 by adding, “In our study’s subsample…”, and this percentage also appears in Table 2.

6. Minor changes

On line 195, add 'gut' between human and microbiome.

Line 222 Mention also that bacteriodes can also cause serious infections if escape outside the gut, potentially leading to septicemia too.

Line 282 The percentage of breastfed varies by age of infant. Do you mean ever breastfed, exclusively breastfed, or breastfed to 6 months? More specificity needed.

Line 292 remove formula; it may be interpreted as 'better than breastfeeding' and that's not the approach supported by WHO or multiple health agencies.

All minor changes noted by the reviewer have been made.

Thank you again for the opportunity to revise our manuscript. 

Sincerely,

---

## [Editor Report · Decision Letter 2]

11 Jun 2020

History of breastfeeding but not mode of delivery shapes the gut microbiome in childhood

PONE-D-20-02497R2

Dear Dr. Leve,

We’re pleased to inform you that your manuscript has been judged scientifically suitable for publication and will be formally accepted for publication once it meets all outstanding technical requirements.

Kind regards,

Corrie Whisner

Academic Editor

PLOS ONE
---

## [Editor Report · Acceptance letter]

23 Jun 2020

PONE-D-20-02497R2 

History of breastfeeding but not mode of delivery shapes the gut microbiome in childhood 

Dear Dr. Leve:

I'm pleased to inform you that your manuscript has been deemed suitable for publication in PLOS ONE. Congratulations! Your manuscript is now with our production department. 

Kind regards, 

on behalf of

Dr. Corrie Whisner 

Academic Editor

PLOS ONE